# The Memory of a Fault Gouge: An Example from the Simplon Fault Zone (Central Alps)

**Valentina Argante** [1,*], **David Colin Tanner** [1], **Christian Brandes** [2], **Christoph von Hagke** [3] and **Sumiko Tsukamoto** [1]

1   Leibniz Institute for Applied Geophysics, 30655 Hannover, Germany; davidcolin.tanner@leibniz-liag.de (D.C.T.); sumiko.tsukamoto@leibniz-liag.de (S.T.)
2   Institut für Geologie, Leibniz Universität Hannover, 30167 Hannover, Germany; brandes@geowi.uni-hannover.de
3   Department of Geography and Geology, University of Salzburg, 5020 Salzburg, Austria; christoph.vonhagke@plus.ac.at
*   Correspondence: valentina.argante@leibniz-liag.de

**Abstract:** Faut gouge forms at the core of the fault as the result of a slip in the upper brittle crust. Therefore, the deformation mechanisms and conditions under which the fault gouge was formed can document the stages of fault movement in the crust. We carried out a microstructural analysis on a fault gouge from a hanging-wall branch fault of the Simplon Fault Zone, a major low-angle normal fault in the European Alps. We use thin-section analysis, together with backscattered electron imaging and X-ray diffractometry (XRD), to show that a multistage history from ductile to brittle deformation within the fault gouge. We argue that this multistage deformation history is the result of continuous exhumation history from high to low temperature, along the Simplon Fault Zone. Because of the predominance of pressure solution and veining, we associated a large part of the deformation in the fault gouge with viscous-frictional behaviour that occurred at the brittle-ductile transition. Phyllosilicates and graphite likely caused fault lubrication that we suggested played a role in localizing slip along this major low-angle normal fault.

**Keywords:** fault gouge; Simplon fault; Alps

## 1. Introduction

Fault gouge is commonly present in the cores of fault zones. Fault gouge is an end member of those fault rocks that develop by cataclastic processes [1,2]. It consists of an incoherent rock with greater than 70% fine-grain material in the rock volume. The fine-grained matrix is the result of incremental deformation, during which there is a continuous process of fracturing, abrasion, rotation, and grain-size reduction [3]. The mineralogical composition of the fault gouge is theoretically the same as the protolithic rock in the absence of any chemical reactions, since its origin is monogenetic. However, fluid-rock interaction can change the mineralogical composition, in particular, by the formation of new phyllosilicates, such as micas and clays.

Laboratory experiments [4–8] and field evidence [9–14] demonstrate that the presence of phyllosilicates causes strain-weakening processes. Bos et al. (2000) [15] also show that the presence of the phyllosilicates causes a frictional-viscous behaviour, a process that also includes pressure solution and cracking under hydrothermal conditions. A wealth of literature is available on the role that phyllosilicates have on the rheology of the fault, in decreasing the strength of the fault itself, and its dip angle. It has been recognized for several decades that the material properties of a gouge have a strong impact on the sliding behaviour of a fault [16]. Generally, argillaceous material in a fault core has strain-softening behaviour [17] and clay-rich fault rocks can promote aseismic creep on faults [18]. Phyllosilicate-rich gouge material shows velocity-strengthening at low velocities (but

velocity weakening at high velocities [19] and sometimes it can also favour fault creep [20]). Thus, the material properties of a gouge can provide valuable information about the dynamic behaviour of a fault. For all these reasons, analysis of gouge microstructures can add to a more complete understanding of the mechanisms of faulting. Nevertheless, because of the transient nature of the fault gouge, it is uncommon to observe all the different stages of microstructure within it.

The Neogene Simplon Fault [21–30] in the Central Alps is one of the most important normal faults that accommodated post-orogenic orogen-parallel extension. Its activity has allowed the exhumation of the lower crust in its footwall, composed of Helvetic and Penninic nappes [26–28,31–36]. The Simplon Fault Zone (SFZ) is an excellent example of extensional detachment [26–28,37,38], where brittle deformation (i.e., Simplon Line) overprints the ductile deformation (Simplon Shear Zone). The SFZ includes fault rocks ranging from ductile mylonites and pseudotachylytes to brittle fault gouges. Along the SFZ, the footwall was rapidly exhumed and cooled from maximum temperatures exceeding 600 °C (>20 km depth) to near-surface conditions of less than 100 °C [26,27,33]. An extensive thermochronological data set from the Central Alps [36] is available and allow the detailed analysis of the exhumation along the SFZ [33,36,39–41].

In this study, we investigate the microstructures and the deformation mechanisms that occurred inside a fault core very close to the SFZ. The microtexture of a coherent portion of the fault gouge, together with data available from the literature, allows us to determine which deformation mechanisms prevailed, and crucially, draw implications about the behaviour of this major detachment. We consider it highly interesting to extrapolate information from the fault gouge, in the light of what is already available from previous studies of this fault zone.

## 2. Geological Setting

### 2.1. Central Alps

The Alps are an orogenic belt that extends from France to the west to Slovenia to the east, approximately 1200 km along strike. Typically, the chain has been divided into three different portions, characterized by different tectonic and/or paleogeographic units and deformation history: Western, Central, and Eastern Alps [42]. The Alpine Orogeny was and still is the result of convergence between the European and African Plate [43–45]. In the western Alps the convergence movement caused, since the Cretaceous, the first subsidence of the Piedmont-Ligurian Ocean [46,47], interposed between the two continental plates, and later (Cenozoic) orogenic collision [48]. The latter caused crustal thickening during continental collision and, in particular, by the subduction of the European passive margin in the Early Eocene. A new post-collisional stage of extension, heralded by normal faults and conjugate strike slip faults, began after the crustal thickening and allowed the formation of tectonic windows that expose the lower crust made of Penninic or European units [42–50]. In the Central Alps, the SFZ is related to this last stage post-collisional extension [25,26,31–39].

### 2.2. Simplon Fault Zone (SFZ)

The Simplon Fault Zone (SFZ) is a major low-angle normal fault that accommodated extension during the post-collisional phase of the Central Alps, beginning in the Oligocene [27]. Thermal modelling, based on a wide range of different thermochronometers, suggests the fault was most active between 18 and 15 Ma, although movement continued at a lesser rate down to at least 5 Ma [33]. Total slip is approximately 36 km, with 6 km of that in the last 15 Ma [33].

The SFZ strikes east-west for 30 km along the Rhône Valley (as the Rhône Line) to the northwest, across the Simplon Pass it strikes SE-NW (as the Simplon Line) to the Ossola Valley to the southeast, where its termination is still under debate [51] (Figure 1a). In the central part (Simplon Line), the fault dips 25–30° to the SW. The SFZ at this location is composed of a mylonite, ca. 2 km thick, overprinted by a narrow cataclastic zone, a few

metres thick (Figure 1b) [27,35,41]. The fault allows the exhumation of deeper rocks of a Lower Pennic unit, part of the Lepontine metamorphic dome, in the footwall, whereas Upper Pennic units occur in its hanging wall. The area surrounding the SFZ is not only deformed by the southwest movement of the fault itself: NW-vergent thrusting, SE backthrusting, and dextral transpression overlap each other in the same area [25,52].

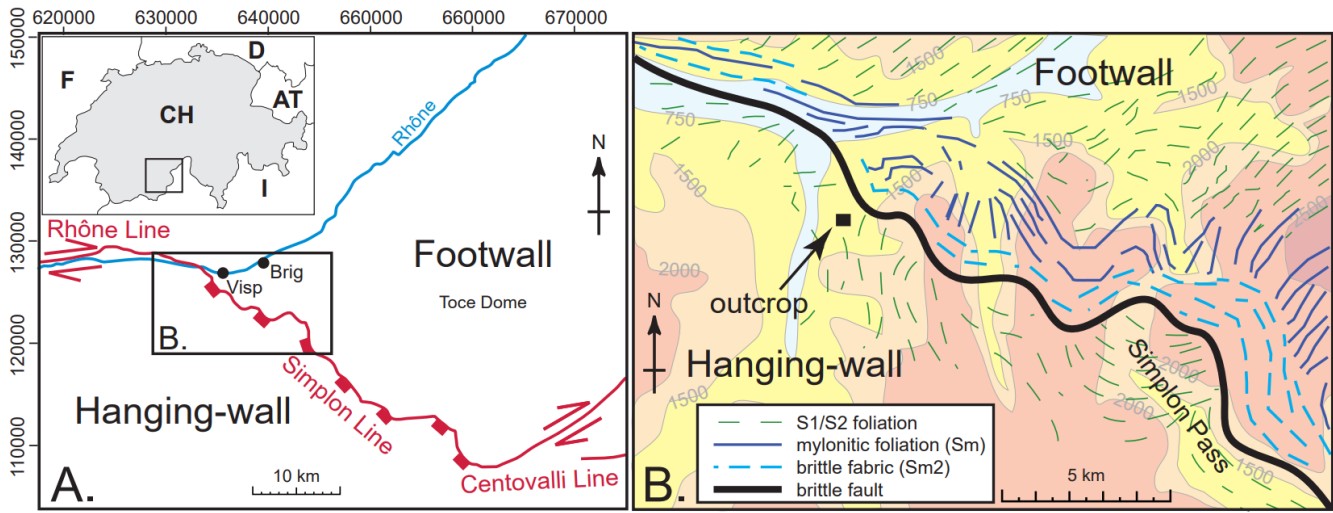

**Figure 1.** Geological overview: (**A**) Major parts of the Simplon Fault Zone and its location in southern Switzerland and northern Italy. Box shows the area in (**B**) Close-up of the Visp area, showing the outcrop location, the orientation of the brittle fault and ductile fabrics of the SFZ, and foliation in the footwall and hanging-wall (after Campani et al. 2010 [36]), underlain by topography, contours at 500 m.

According to [35], kinematic indicators from both the ductile and brittle components show a top-to-the-SW shear sense, in contrast to older structures preserved in the hanging-wall. The same authors also noticed that ductile structures are non-pervasively overprinted by brittle faults.

## 3. Methods

The fieldwork for this study was carried out in early August 2019. The outcrop is located approx. 1 km south of Visp (Canton Wallis), directly at the roadside between Visp and Unterstalden. It is situated in the hanging wall of the Simplon Fault Zone, very close (<200 m) to the main fault (Figure 1). Previous work on this same outcrop was carried out by [35].

In the field, we took a number of samples from the core of the fault zone, where soft material occurs together with lenses of more solid material and material that is cemented by quartz (thin sections were made from the solid material). Furthermore, we made measurements of the orientation of the different slip surfaces, and documented the outcrop photographically.

To study the microstructures that occur in the fault gouge we analysed thin sections under an optical microscope and by backscattered electron imaging (BSE) and energy dispersive X-ray spectroscopy (EDX) using a scanning electron microscope (SEM, Sigma 300 VP). We worked on six oriented thin-sections, on two different planes: one parallel to the slip movement and another one perpendicular to it.

To define the mineralogical composition of the major mineral phases, we used XRD analysis. The XRD patterns were recorded using a PANalytical X'Pert PRO MPD Θ-Θ diffractometer (CuK radiation generated at 40 kV and 40 mA), equipped with a variable divergence slit (20 mm irradiated length), primary and secondary soller, Scientific X'Celerator detector (active length 0.59°), and a sample changer (sample diameter 28 mm). The samples

were investigated from 2° to 85° 2Θ with a step size of 0.0167° 2Θ and a total measuring time of 48 min. For specimen preparation, the back-loading technique was used.

## 4. Results

### 4.1. Fieldwork and Outcrop-Scale Structures

The investigated outcrop is localized in the hanging-wall of the mapped SFZ. The dip angle of the fault, around 50–60°, is higher than the average 25–30° dip of the SFZ and suggests the fault is a branch fault that formed to accommodate the internal deformation and extension in the hanging-wall within the SFZ. The main element of the outcrop is a fault zone with a distinct high-strain area (ca. 1.5 m thick), in which the slip along the fault was manifested and thus can be classified as a fault core (Figure 2) [53,54]. The material in the fault core can be classified as gouge and partially also as fault breccia, because of the high clast contents [2,55]. The core shows material with high strain fabrics, which is incoherent and with an abundant and variable quantity of fine matrix. Within this soft material, solid and foliated lenses (centimetre to decimetre size with a phacoid or sigmoidal shape, Figure 2B), were conserved, due to the quartz cement that preserved fabrics, which otherwise would have been destroyed by the cataclastic processes. In some parts of the outcrop, these lenses dominate and some parts of the fault core consist mainly of the crushed material. The outcrop is dominated by phyllosilicates that form the main fabric (foliation) in the core.

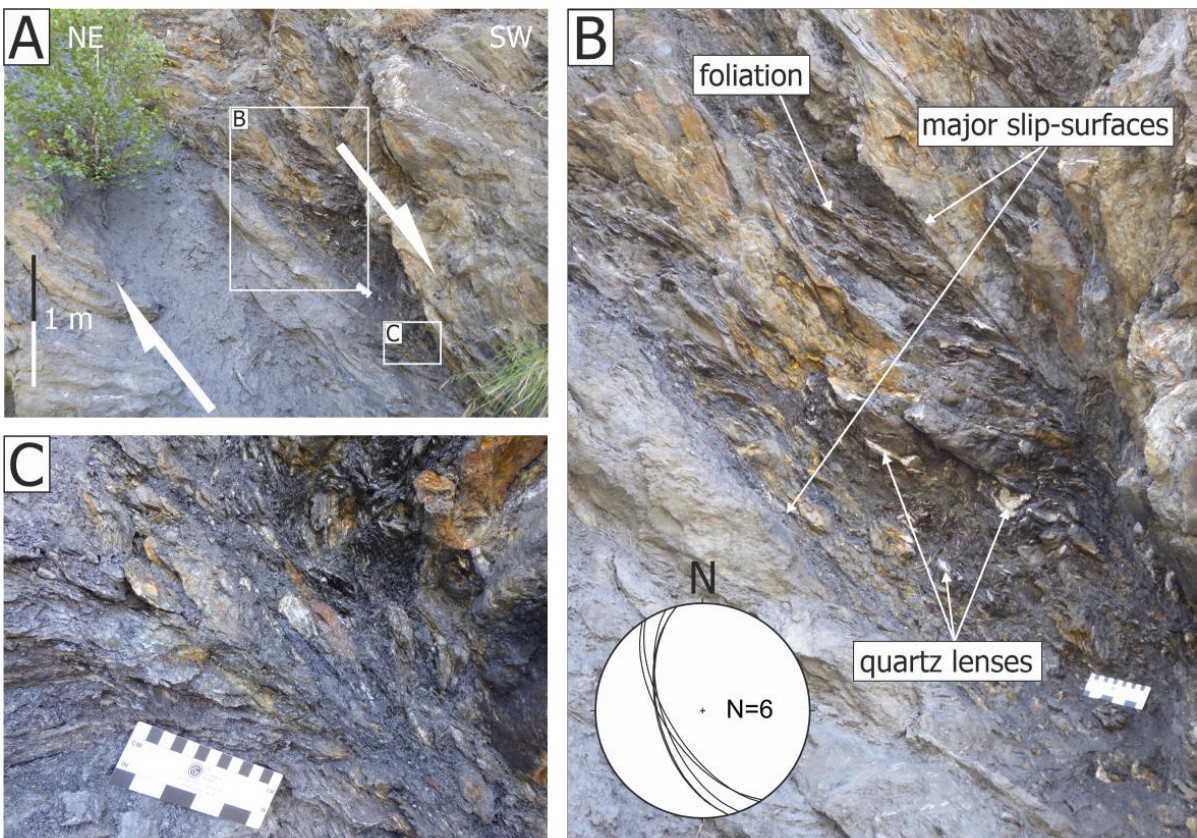

**Figure 2.** Photographs of the outcrop. (**A**) Picture of the outcrop, showing the thickness of the fault gouge and the macrostructure; (**B**) detail of the internal foliation (insert: stereographic projection of the slip planes); (**C**) the contact of the internal foliation with the wall of the fault core.

The core foliation and the slip surfaces meet at an acute angle (Figure 2B). We interpret this as a Riedel shear pattern, where the new, shallowly-dipping, spaced foliation is oriented parallel to the P-surfaces (Figure 3A, 47°), the slip surfaces (60°) are then R1- and Y-surfaces. This is consistent with normal, dipping to the west, kinematics (Figure 3).

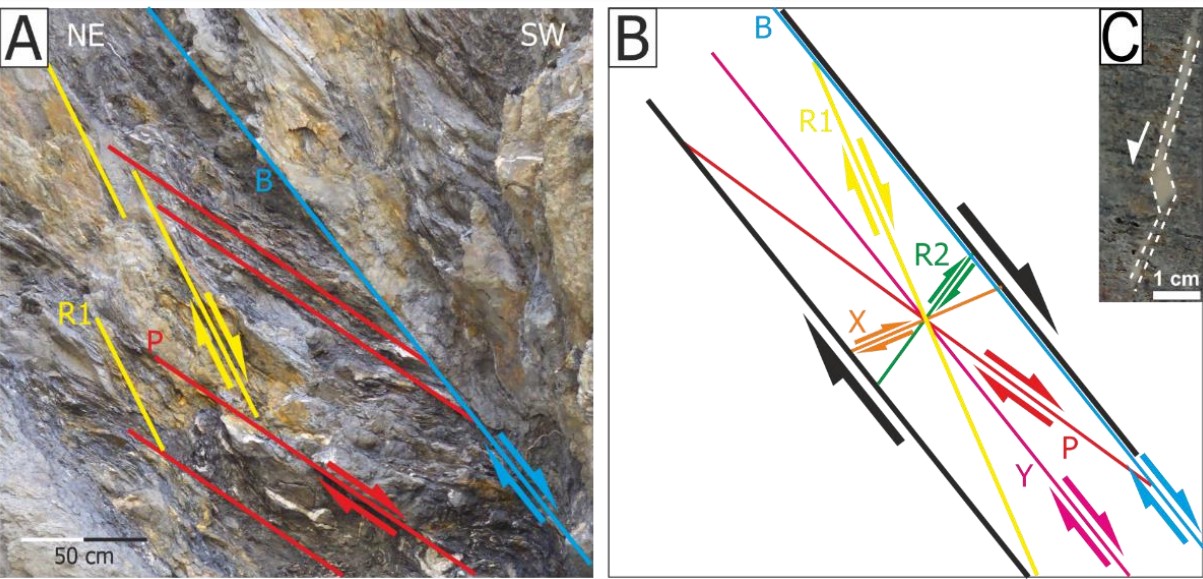

**Figure 3.** Kinematic interpretation of fabrics of the fault core. (**A**) Riedel planes inside the fault core; (**B**) schematic pattern of Riedel shear planes that usually develop in a brittle shear zone; modified after Ikari et al. 2011 [56]. (**C**) Extensional jog, with antithetic kinematics (parallel to R2), seen in the hand sample.

## 4.2. Microstructural Analysis

In terms of minerals, the fault core consists of, from higher to lower dominance: quartz, muscovite, illite, chlorite, calcite, and albite. Other minerals present in smaller quantities are rutile and k-feldspar (Figures 4 and 5). We are also able to identify opaque minerals such as graphite c.f. [48–57].

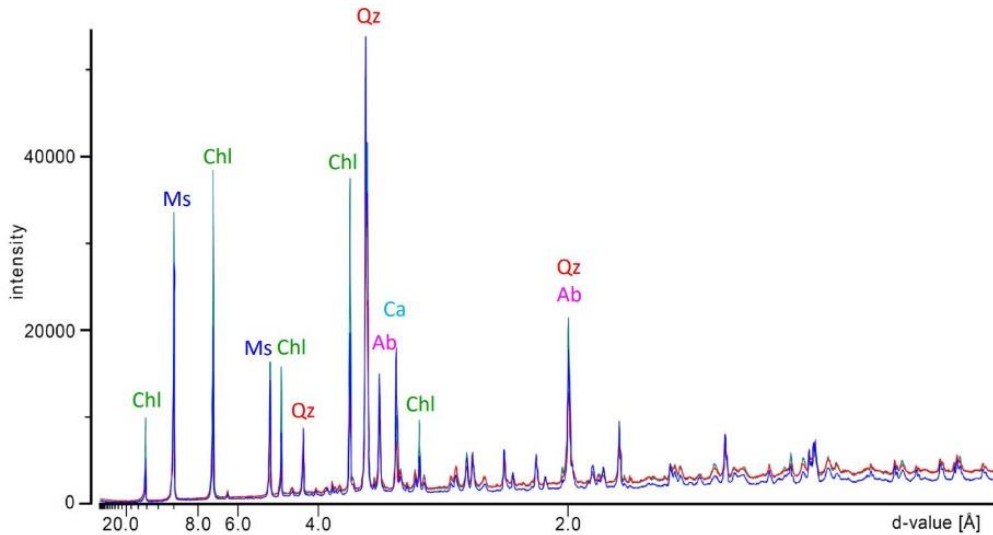

**Figure 4.** XRD analysis of the fault core sample. The major reflections are caused by quartz, chlorite, albite, calcite, rutile, and muscovite. Three different samples were analysed, shown as blue, red, and green lines.

In thin-section, we observe a distinct foliation, comparable to the foliation observable at the mesoscale and macroscale (see fieldwork observations): it consists of muscovite-illite and chlorite (Figure 6a,e,f), mixed with graphite. This planar structure can be identified in the sections, both parallel and perpendicular to the slip of the fault (Figures 6f and 7S2).

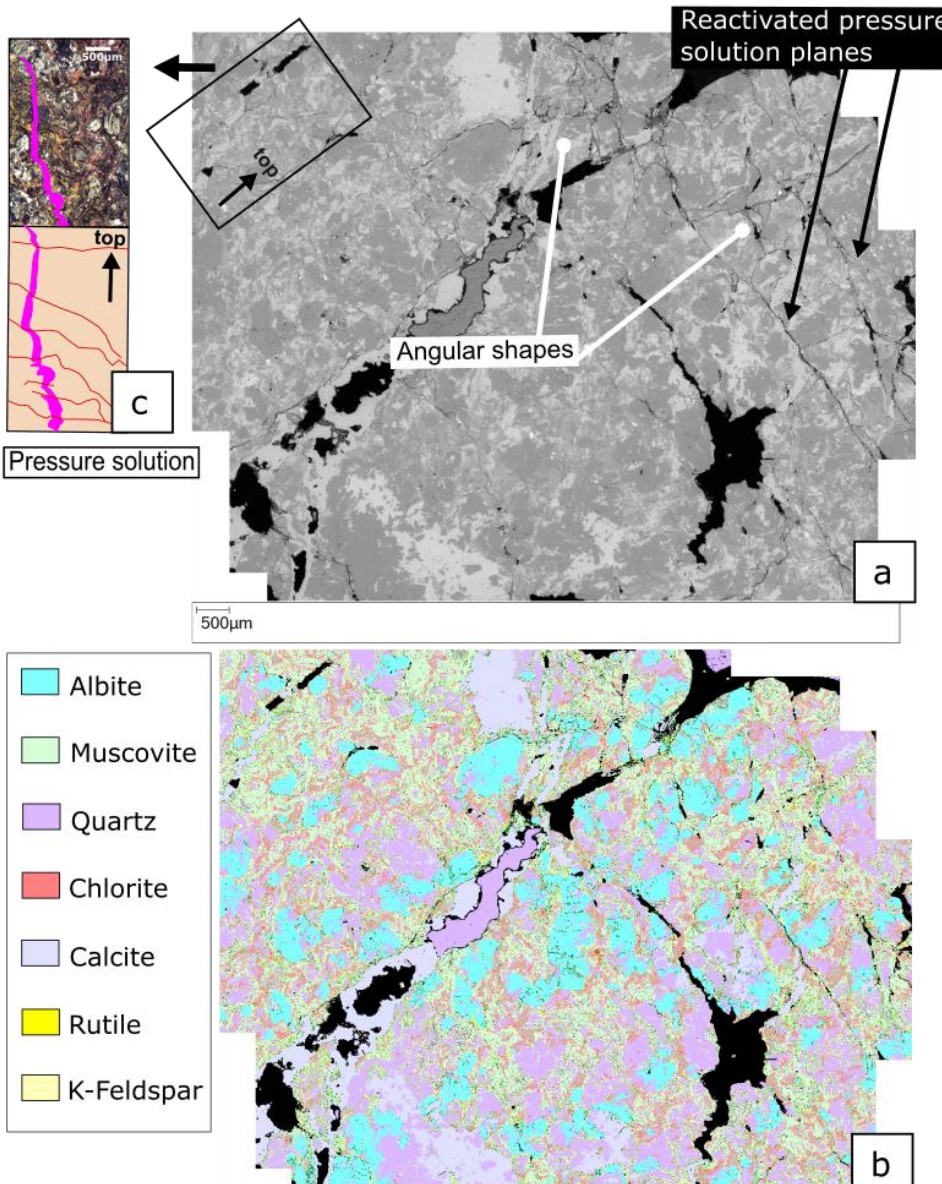

**Figure 5.** (**a**) Backscattered electron-scanning microscopy (BSE SEM) imaging, of a thin section perpendicular to the fault dip. The major mineralogical composition is colour coded in figure (**b**) (image of the EDX mapping). Picture in (**c**) shows a comparison of the BSE-SEM image of the same portion under cross-polarized light, showing pressure solution affecting a vertical calcite vein.

We recognize folded and stretched quartz veins that cross the foliation (Figure 7S1). The fold axial planes of the quartz veins are, when preserved, perpendicular to the main foliation. These deformation structures, evidently linked to a distinct deformation phase (D1), also involved the redistribution of muscovite: muscovite surrounding the folded quartz is perpendicular to the main foliation (i.e., parallel to the fold axial planes), in contrast with the most pervasive deformation (Figure 7f). This texture is found mainly in the sections perpendicular to the slip direction of the fault. In the thin section, the quartz is strongly stretched along the foliation, parallel to the slip direction.

The highest grade of deformation is indicated by quartz grains in the veins. Dynamic recrystallization is present widespread in quartz as grain-boundary migration/sub grain rotation (GBM-SGR), bulging (BLG) mechanisms, and deformation lamellae (Figure 8). We term this high-temperature deformation D1 (i.e., GBM-SGR) and low to medium T deformation D2 (BLG and deformation lamellae).

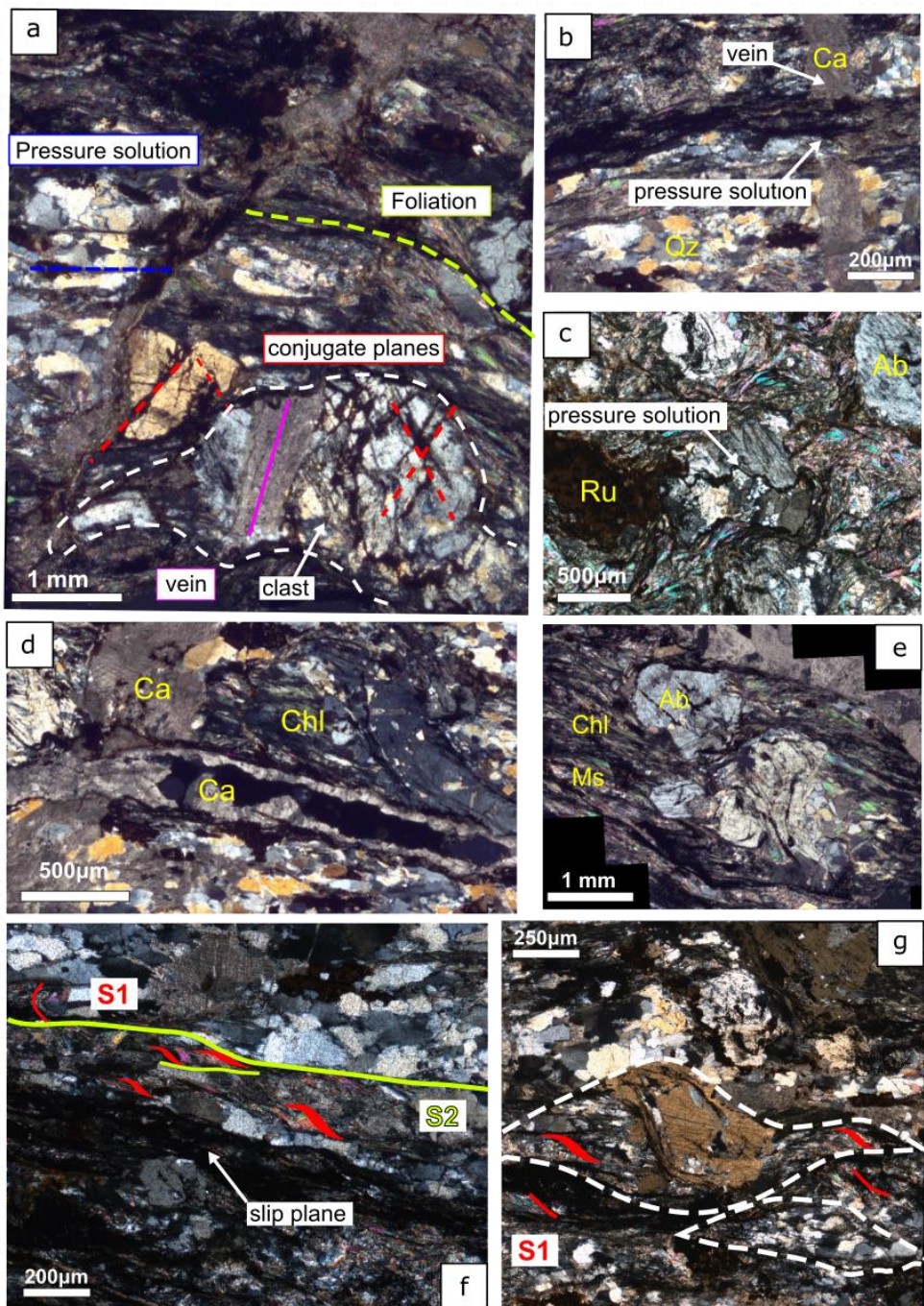

**Figure 6.** Thin sections under cross-polarized light; (**a**) D3 foliation of the fault core (blue line), with high angle fractures (red) and pressure solution planes (in yellow) from which the orientation of the paleostress field, with a near-vertical maximum principal axis, can be derived. A composite clast is outlined, containing a fragment of a calcite vein; (**b**) pressure-solution vein cutting a calcite vein; (**c**) detail of the pressure solution affecting two composite clasts: in the lower individual minerals can still be observed; (**d**) cross-cutting relationship between two calcite veins; the horizontal vein probably used an older pressure-solution plane filled with chlorite; (**e**) Euhedral feldspars within intense foliation parallel to the orientation of the fault core. (**f**) previous S1 foliation deformed by the younger S2 foliation (see Campani et al. 2010 [35] and Montemagni and Zanchetta 2022 [41] for similar features) and the slip plane developed along a pressure solution plane; (**g**) sigmoidal clasts made of feldspar and quartz, deformed by the younger S1 foliation and shear deformation along S1. Within the clast, an inter-tectonic foliation can be identified.

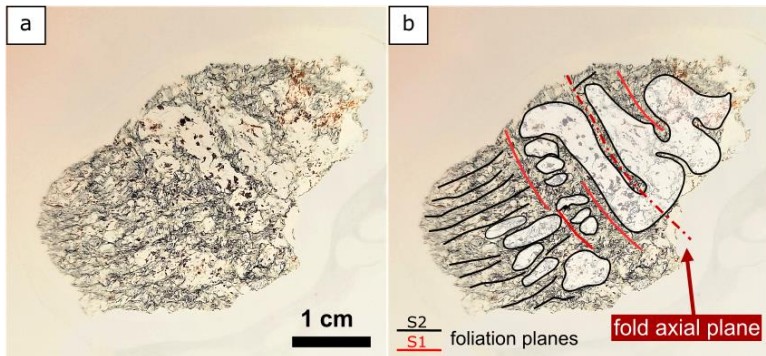

**Figure 7.** Thin section perpendicular to the fault slip, showing a folded quartz vein. (**a**) Uninterpreted photo; (**b**) Interpreted photo showing the two different foliations, S1 and S2. The quartz vein is partially overprinted by the younger S2 foliation (during D2, black lines), perpendicular to the fold axial planes (S1/D1, red lines).

Calcite veins are homogeneously distributed throughout the rock, oriented mainly perpendicular to the main foliation. However, several generations of veins can be identified: some are almost vertical or horizontal oriented, occasionally they fill the porosity of the rock; rare veins are present in albite crystals or within composite clasts (Figure 6a). The same albite can be found with euhedral forms as porphyroclasts, with internal previous deformation fabric (Figure 6e).

The dominant deformation mechanism in the sample is pressure solution (D3). We also recognize that sub-vertical calcite veining occurred simultaneously to pressure solution. Pressure solution affected mostly quartz grains; albite reacted in a more brittle fashion and only some fractures are locally filled by calcite. Pressure solution is also present around composite clasts (Figures 5c and 6c).

In addition to the calcite veining, other brittle structures can be found (D4), such as micro-fractures at crystalline scale and extensional jogs with shear movements, which also suggest the normal kinematics of the sample (Figure 3C). Of particular importance are micro-fractures along with certain crystallographic directions of the quartz: under polarized light the small distinct subgrains do not have different extinction, confirming that they are not subgrains, but an expression of brittle deformation (Figure 8c). These processes cause the breakdown of the quartz grains, forming zones of cataclasis (Figure 8a). This kind of feature is typical of even impact structures and some authors, such as Poelchau and Kenkmann 2011 [58], have attributed them to the shock waves of earthquakes.

In summary, the microstructures recognized in the fault core samples are attributable to deformation under both ductile and brittle conditions. The microstructures can be linked to deformation mechanisms that occurred at different pressures and temperatures that occurred when the fault rock passed through different depth levels before reaching the surface. For instance, the GBM deformation in the quartz formed under high temperature and dynamic conditions (T ~500–600 °C); it also affected the quartz in the folded veins. GBM occurs via diffusion creep under ductile conditions [59,60]. The presence of chlorite and muscovite, as well as the BLG mechanism in quartz, suggests that the rock was later exposed to lower temperature and pressure conditions (greenschist facies, T = 300–400 °C, P = 0.3–0.5 GPa). The chlorite is only relictic, which suggests that even lower T–P conditions were reached during the development of the SFZ. We also recognize pressure-solution deformation to be coeval with calcite veining, which is evidence of the large amounts of fluid precipitation at this stage. Calcite occurs as vertical (perpendicular to the foliation) and conjugate fractures that indicate a sub-vertical maximum principal stress, as one would expect in the hanging-wall of a normal fault. Calcite, however, also precipitated along discontinuities, such as S2 foliation planes.

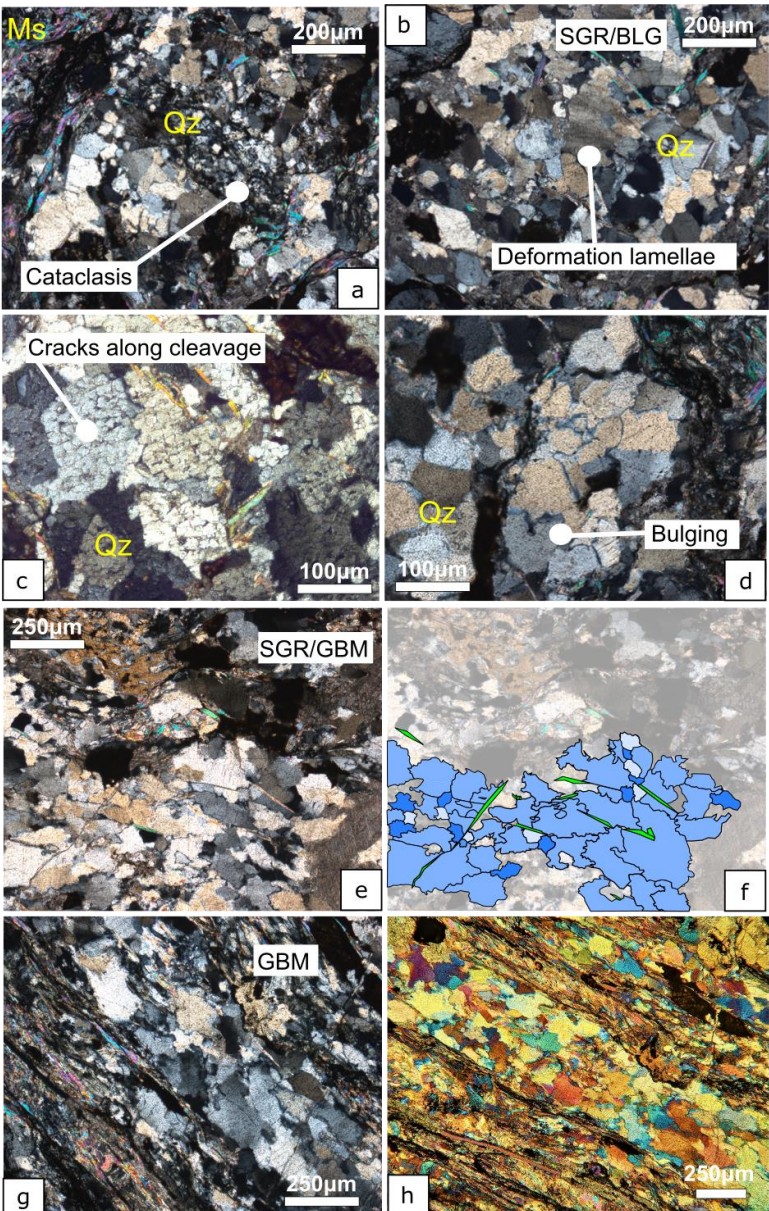

**Figure 8.** Thin sections under crossed-polarized light; (**a**) cataclastic breaking of quartz parallel to the fault core foliation; Evidence of deformation lamellae (**b**), brittle deformation inside quartz (**c**), bulging (**b**,**d**) and grain boundary migration (**e**–**h**). (**e**,**f**) are the same pictures in which we highlight the shape of the quartz boundaries in (**f**), as well as the formation of new subgrains with an average size of 150 μm. (**g**,**h**) are the same portion of the thin section under cross-polarized light (**g**) and with lambda plate inserted (**h**).

## 5. Discussion

### 5.1. Macrostructure of the Gouge

Faults can show a great variety of slip behaviour, ranging from aseismic creep, over slow-slip events, seismogenic rupturing to super-shear slip [54]. The fabric developed in a gouge zone is a function of fault movement and the metamorphic environment [61]. In turn, the fault core material directly influences the fault slip-processes [16,62,63]. Thus, analyzing the internal structure and material properties of a gouge zone is crucial to understanding earthquake nucleation and evolution [64,65]. The studied gouge material shows a shear-related foliation and fracture pattern with a systematic orientation related to the general shear sense (Figure 7). A similar pattern has been observed in natural [66,67]

and experimental gouges [10–68]. The presence of a well-developed fabric in the gouge might advocate for the large accumulation of slip and high strain along this branch of the SFZ. This concurs with experimental studies of Marone et al. (1990) [69] who show a progressive evolution from Riedel shears to fully-developed boundary shears (B-shears) with increasing strain, as well as with the work of Scuderi et al. (2017) [70], which shows that with increasing strain, B-shears become more continuous. Especially the Riedel shears in the gouge are important, as they are interpreted as indicative of seismogenic slip [71] and thus point to stick-slip behaviour along this analysed branch of the SFZ. This might indicate that the youngest movements along this fault were accompanied by earthquakes and make the outcrop a promising target for further geochronological studies that can date the last fault activity with a frictional-heat based approach, such as ESR-dating.

### 5.2. Exhumation History

Cataclasis is a process typical of the upper crust, which commonly takes place above the brittle-ductile transition, which for quartz is located where the temperature reaches approximately 300 °C. In an orogenic zone such as that of the SFZ, this occurs at a depth of 15–20 km [72]. Typically, a mixture of clasts and matrix characterizes the core of brittle faults. However, in the case of the Simplon Line, the evolution is more complex. The Simplon Line affected the rocks of a shear zone that developed 35 Ma ago. This means that the protolith of the brittle fault core was most likely a mylonite of the Simplon shear zone, which was later overprinted under brittle conditions. This hypothesis is confirmed by our analysis of the texture of the fault rock. The samples show a very complex fabric and thus cannot be simply defined as a fault breccia or fault gouge, although the SEM-BSE images (Figure 8) show that fractures and rotating angular structures are present in the sample. Moreover, both typical cataclasitic and mylonitic characteristics are not well recognizable. The co-presence of clasts that roll and shatter along the foliation, as well as sigmoidal structures and the ribbon quartz with deformation mechanisms relative to temperatures relative to ductile regimes could be identified. We infer that this set of structures is typical of fault rocks, in which continuous deformation involves a pre-existing mylonitic shear-zone that is later incrementally overprinted under brittle conditions during the exhumation process. We can also explain the co-presence of ductile and cataclastic features as the consequence of the oscillations that occur in correspondence with the brittle and ductile conditions. We assume that the D1 characteristics are preserved features of the mylonite (the protolith of the brittle fault zone). It is likely that the texture of the quartz made these lenses more competent in the comminution process in the fault core. Unfortunately, the typical cataclasitic fabric was obliterated by the clear prevalence of phyllosilicates, which accommodated the slip and reduced the friction and abrasion during fault movement.

In Figure 9, we summarize the fabrics of the fault core sample. In the quartz veins and fragments of them, the oldest deformation of the samples is recorded (D1; recrystallization due to GBM, T ~ 500–600 °C), together with a low-grade metamorphism (D2), both related to the protolithic rock paragneiss [35,36]. The D1 phase is preserved thanks to the folding, which has preserved earlier fabrics in this part of the sample. In all other parts, the S2 foliation is dominant and supports slip along the fault, probably also reactivating pressure-solution planes: phyllosilicate and pressure solution planes have been reactivated as brittle slip planes of the fault (D3). In addition, the high pressure of fluids sometimes caused calcite veins to be injected parallel to the slip planes. Pressure solution is typical at depths and temperatures at which quartz undergoes plastic behavior, suggesting the samples were exhumed from around the brittle-ductile transition. 'Cleavage cracking' and cataclasis of quartz confirm this, since they can be also formed by shock waves of earthquakes that occur close to the transition zone. Nevertheless, they can also be formed during the exhumation; Kimberly et al. (2010) [73] experimentally prove that tensile cracks can form under the unloading following high compression. Either way, during exhumation the analyzed fault rock was brought to the surface, a process that involved cataclasis sensu stricto and lead to the production of a fine and incoherent fault gouge (D4).

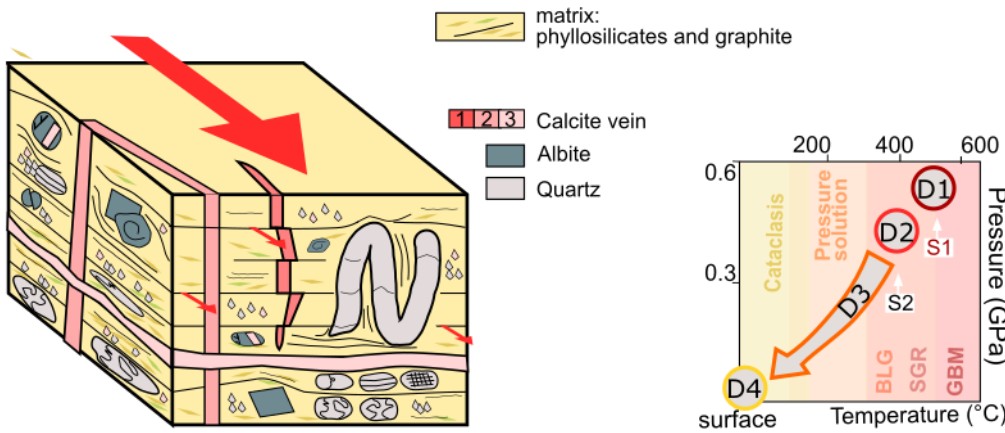

**Figure 9.** Left; a 3-D block cartoon to show the deformation features. Grey—remnants of folded quartz veins (D1), pink—calcite veining (D4), black lines—D2 foliation and D3 pressure solution planes, green minerals—albite grains. Right; a P-T diagram to summarize the different stages of the fault gouge sample during exhumation (supported by Rings & Merle 1992 [39]; Grasemann and Mancktelow 1993 [33]; Campani et al. 2010 [36]; Bousquet et al., 2002 [74]; Montemagni and Zanchetta [41]; Cawood and Platt 2000 [75]). BLG—grain bulging, SGR—sub-grain rotation, and GBM—grain boundary migration.

### 5.3. Fault Lubrication

Our analysis of the analysed gouge material shows that, besides the dominant quartz and feldspar minerals, the gouge also contains phyllosilicates, muscovite, and chlorite (Figures 2–4). Friction experiments show that the presence of chlorite causes the rock to weaken. Friction decreases nearly linearly with increasing chlorite content, thus implying that retrograde chlorite growth leads to frictional weakening and thus stable slip in higher-grade mineral assemblages exhumed to lower temperature conditions [67]. It is likely that a similar process took place along the SFZ. Foliated fault rocks have a lower frictional strength in general and sliding processes on the foliation planes control the reduction in fault strength [76]. The presence of a foliation in the outcrop and the thin-sections supports the interpretation of a weak fault core in this part of the SFZ.

The thin-section analysis reveals that the analysed gouge also contains a certain amount of graphite. Graphite in gouges is not uncommon [27,77–79], and gouges can be composed of crushed quartz and feldspar fragments together with highly crystallized graphite [80]. Similar to phyllosilicates, graphite can also have a lubrication effect, since it is a material with very low friction [63–81]. Experimental studies on mixed graphite-quartz gouges indicate strain hardening at low shear strains and/or strain weakening with increasing deformation [82]. Thus, the presence of graphite in the analysed gouge suggests it lowered friction. The enrichment of graphite along fault slip zones is seen as an indicator of transient frictional heating [83] and is a reliable tracer of seismic slip [84], thus implying that the gouge material analysed in this study was likely once located in a seismogenic zone, as shown by the exhumation history. Fault-weakening mechanisms are regarded as crucial to allow slip along faults that are not optimally oriented, where a low resolved shear stress on the fault surface in combination with highly effective normal stress occurs. Both the phyllosilicates and the graphite present in the analysed gouge material could have had a significant effect on the frictional behaviour of the SFZ. Furthermore, the combination of pressure-solution, the presence of phyllosilicates, and the hydrothermal conditions at the brittle-ductile transition zone is the same as experimentally derived by Bos et al. 2000 [15], for a viscous-frictional fault behaviour. Lastly, fault-weakening due to low-friction gouge material is a possible way to explain movement on low-angle normal faults and thus might have played a role in the development of the Simplon fault, as proposed for other low-angle continental normal faults [68].

## 6. Conclusions

The Simplon Fault Zone is a major fault of the Alpine chain in which the current state of exhumation allows us to observe ductile mylonite, as well as brittle fabrics made of cataclastics and fault gouge. Studying the deformation mechanisms of which fault rocks form can give valuable information about the dynamics behaviour of a fault (seismogenic vs. creep), which has significant implications for understanding fault behaviour. Our main conclusions are:

1. The dominant deformation mechanism is pressure-solution creep.
2. Pressure-solution creep overprinted earlier ductile fabrics. This, together with the involvement of fluids and phyllosilicates, suggest the rock passed through the brittle-ductile transition at this time.
3. We recognise cleavage cracks that can be linked to the seismicity of the fault.
4. The formation of the pressure-solution foliation, formed by muscovite, chlorite, and graphite, significantly reduced friction. If a similar activity occurred on the SFZ, it would explain why the normal fault of the SFZ has such a low, otherwise unfavourable angle.

In conclusion, we find a protracted deformation history of the SFZ witnessed in a variety of brittle and ductile structures. Interestingly, brittle-ductile structures are most prevalent, indicating the studied section of the fault was at depths of 15-20 km for a long time before being relatively quickly exhumed to purely brittle conditions. This notion is supported by previous studies, showing late stage rapid exhumation along the Simplon Fault. However, future studies applying e.g. ESR thermochronometry would be required to determine the time evolution of exhumation more precisely.

**Author Contributions:** Conceptualization, D.C.T. and C.B.; validation, S.T., D.C.T., C.B. and C.v.H.; formal analysis, V.A.; investigation, D.C.T., C.B. and V.A.; resources, S.T.; data curation, V.A., D.C.T. and C.B.; writing—original draft preparation, V.A., D.C.T. and C.B.; writing—review and editing, S.T., D.C.T., C.B. and C.v.H.; visualization, S.T., D.C.T., C.B. and C.v.H.; supervision, S.T. and D.C.T.; project administration, S.T. and D.C.T.; funding acquisition, S.T., D.C.T., C.B. and C.v.H. All authors have read and agreed to the published version of the manuscript.

**Funding:** This research was funded by German Science Foundation (DFG), grant number 442589783 for D.C.T., C.B., C.v.H. and S.T.

**Institutional Review Board Statement:** Not applicable.

**Informed Consent Statement:** Not applicable.

**Data Availability Statement:** Not applicable.

**Acknowledgments:** We acknowledge the help of Reiner Dohrmann, Kristian Ufer, and Niko Götze to carry out the XRD measurements. Andre Marx and Matthias Halisch are thanked for their help in the laboratory.

**Conflicts of Interest:** The authors declare no conflict of interest. The funders had no role in the design of the study; in the collection, analyses, or interpretation of data; in the writing of the manuscript, or in the decision to publish the results.

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
