# Peer review of "The Memory of a Fault Gouge: An Example from the Simplon Fault Zone (Central Alps)"

_geosciences, doi:10.3390/geosciences12070268_

Round 1

Reviewer 1 Report

The authors try to investigate the microstructures and the deformation mechanisms that occurred inside a fault gouge very close to the SFZ. The manuscript is an important subject to elucidate fault history, but sufficient evidence has not been presented. It is recommended to resubmit after the data related to the following comments are supplemented.

  1. Authors should provide a clear field description of the affected fault. For example, the authors said that the target sample was a false gouge, but the photos in Figures 2, 6, and 7 are difficult to define as fault goue. Accurate description of the target sample and the basis for classification should be presented. In other words, please provide accurate evidence whether the target sample corresponds to fault gouge, cataclasite, or mylonite.
  2. The authors used polarized microscope observation as a research method and presented the images as evidence of tomographic history. However, the polarized microscope image alone is insufficient as evidence to support the author's claim. It is absolutely necessary to present SEM-BSE observation images as the polished section target.
  3. The introduction of the paper is citing a number of papers that are not appropriate to present the main purpose of the paper. Please rewrite it with necessary content to fit the thesis topic. 
  4. For the XRD data in Figure 4, it is recommended to redraw the figure format referring to published paper figures. The method of expressing mineral identification is not a general thesis-style picture.
  5. Overall, there is insufficient evidence to derive the fault history of Figure 8. More specific microtextural evidence should be presented systematically.

Author Response

Dear reviewer,

I read carefully your comments about the manuscript we proposed, titled “The Memory of a Fault Gouge: an Example from the Simplon Fault Zone (Central Alps)”. I appreciated your comments and I hope that the new contributes have improved the manuscript.

Point 1:

Authors should provide a clear field description of the affected fault. For example, the authors said that the target sample was a fault gouge, but the photos in Figures 2, 6, and 7 are difficult to define as fault gouge. Accurate description of the target sample and the basis for classification should be presented. In other words, please provide accurate evidence whether the target sample corresponds to fault gouge, cataclasite, or mylonite.

Response 1: We sampled the core of the fault, where soft material occurs together with lenses of solid material and material that is cemented by quartz. This can be clearly seen on the photos that are already in the manuscript. In the text, we have replaced “gouge” with “core” where “fault core” is more appropriate. More explanations about the material have been added in section 4.1.

Point 2: The authors used polarized microscope observation as a research method and presented the images as evidence of tomographic history. However, the polarized microscope image alone is insufficient as evidence to support the author's claim. It is absolutely necessary to present SEM-BSE observation images as the polished section target.

Response 2: A SEM-BSE image of a thin section has been added (Figure 5), in order to confirm our hypothesis

Point 3: The introduction of the paper is citing a number of papers that are not appropriate to present the main purpose of the paper. Please rewrite it with necessary content to fit the thesis topic.

Response 3: More information about the Simplon Fault and exhumation in Central Alps have been included in the latter half of the introduction.

Point 4: For the XRD data in Figure 4, it is recommended to redraw the figure format referring to published paper figures. The method of expressing mineral identification is not a general thesis-style picture.

Response 4: We modified Figure 4 as suggested.

Point 5: Overall, there is insufficient evidence to derive the fault history of Figure 8. More specific microtextural evidence should be presented systematically.

Response 5: More thin section photos have been added in Fig. 5, Fig. 6, Fig.8.

Reviewer 2 Report

This is a very serious study with clear methods and objectives, and I believe its conclusions are supported by its scientific content. I am therefore supportive of its acceptance in Geoscience in its present form.

Author Response

Dear reviewer,

regarding your comment:

"This is a very serious study with clear methods and objectives, and I believe its conclusions are supported by its scientific content. I am therefore supportive of its acceptance in Geoscience in its present form"

we appreciate your interest and your comment about the manuscript.

Thank you from all of us

Reviewer 3 Report

I carefully read the paper entitled “The Memory of a Fault Gouge: an Example from the Simplon Fault Zone (Central Alps)” by Valentina Argante et al.

The paper deals about the Simplon Fault Zone and the fault gouge developed in the hanging wall of the SFZ. The Authors provide some thin-section analysis and X-ray diffractometry. The aim of this research is to show a multistage history from ductile to brittle deformation to constrain the dynamics behavior of the SFZ, together with a continuous exhumation history from high to low temperature.

I feel that the paper must be improved. I listed below major critical points that I see in the paper.

Methods are no way sufficient to support conclusions and exhumation model, for which more references have to be cited and compared.

The provided documentation of images must be improved: better photos of outcrops have to be reported. For figures 5, 6 and 7 please consider to take new shots for them at different light conditions as they are low-quality. Better photos are needed to clearly support that in Fig. 7c and d the quartz is recrystallized via GBM (it does not seem GBM recrystallization mechanism in this photos). You also must show SGR as you based your D1-D2-D3 stages of exhumation model on quartz recrystallization mechanisms.

A wider discussion of structural data in the text is needed to support the model and the conclusion of the Authors.

The referencing is lacking, a wider comparison with already published data on Simplon Fault Zone is necessary.

Other comments are also indicated in attached pdf.

I hope my comments could be useful.

Author Response

Dear reviewer,

I read carefully your comments about the manuscript we proposed, titled “The Memory of a Fault Gouge: an Example from the Simplon Fault Zone (Central Alps)”. I appreciated your comments and I hope that the new contributes have improved the manuscript.

Point 1-4:

Methods are no way sufficient to support conclusions and exhumation model, for which more references have to be cited and compared.

The referencing is lacking, a wider comparison with already published data on Simplon Fault Zone is necessary.

Response 1-4: We have added appropriate references for the model and Simplon Fault.

Point 2: The provided documentation of images must be improved: better photos of outcrops have to be reported. For figures 5, 6 and 7 please consider to take new shots for them at different light conditions as they are low-quality. Better photos are needed to clearly support that in Fig. 7c and d the quartz is recrystallized via GBM (it does not seem GBM recrystallization mechanism in this photos). You also must show SGR as you based your D1-D2-D3 stages of exhumation model on quartz recrystallization mechanisms.

Response 2: Unfortunately, the condition of the fault rocks as well as of the thin sections, make us not able to find for each mechanism a textbook picture to prove them, even if we compare and we found confirm of our observations in literature such us Cawood and Platt 2020 and Montemagni and Zanchetta 2022. Nevertheless new photos have been added to the manuscript (Figure 5, 6 and 8) in order make our observation clearer.

In the new photos include the SGR-GBM features, emphasizing the jagged / lobed edges of the quartz crystals and the new subgrain formation with an average size of 150 µm.

Point 3: A wider discussion of structural data in the text is needed to support the model and the conclusion of the Authors.

Response 3: Discussion (section 5.2) has been rewritten to better support our model.

Other small corrections are made following the suggestions in the pdf file you kindly attached.

Round 2

Reviewer 1 Report

Many previous comments have been reflected and corrected. However, it is not judged that the evidence of the assertion is sufficient. Nevertheless, it is judged that the attempt to interpret the formation process based on the microstructure of fault rocks is meaningful. Therefore, it is recommended that this mauscript be published in Geosciences.

Author Response

Dear reviewer,

We agree to be clearer in the introduction and in the conclusion that the aim of our manuscript is to highlight the potential of the studying of the fault gouge microstructures as a tool, therefore not sufficient itself without other methods and the existing literature.

Thank you for increasing the scientific value of the manuscript.

Reviewer 3 Report

Many previous comments have been clarified. However, in my opinion, methods can not fully support the model. Nevertheless, the attempt to solve a complex deformation history can be potentially satisfying. This manuscript can be relevant to the readers facing with fault gouge, therefore it can be published after correction of some minor typo in the text.

Author Response

Dear reviewer,

In the introduction and in the conclusion we tried to clarify the aim of our manuscript, which is highlighting the potential of the fault gouge analysis as a tool, but not sufficient itself without other methods and the existing literature.

Thank you for increasing the scientific value of the manuscript.
